# pyjeo: A Python Package for the Analysis of Geospatial Data

**Pieter Kempeneers** [1,*], **Ondrej Pesek** [2,†], **Davide De Marchi** [1] and **Pierre Soille** [1]

1   Joint Research Centre of the European Commission, Via E. Fermi 2749, I-21027 Ispra, Italy;
    davide.de-marchi@ec.europa.eu (D.D.M.); pierre.soille@ec.europa.eu (P.S.)
2   Department of Geomatics, Czech Faculty of Civil Engineering, Technical University in Prague, CZ-16629
    Prague, Czech Republic; ondrej.pesek@ext.ec.europa.eu
*   Correspondence: pieter.kempeneers@ec.europa.eu
†   O. Pesek contributed to this work during his traineeship at JRC in 2018.

**Abstract:** A new Python package, pyjeo, that deals with the analysis of geospatial data has been created by the Joint Research Centre (JRC). Adopting the principles of open science, the JRC strives for transparency and reproducibility of results. In this view, it has been decided to release pyjeo as free and open software. This paper describes the design of pyjeo and how its underlying C/C++ library was ported to Python. Strengths and limitations of the design choices are discussed. In particular, the data model that allows the generation of on-the-fly data cubes is of importance. Two uses cases illustrate how pyjeo can contribute to open science. The first is an example of large-scale processing, where pyjeo was used to create a global composite of Sentinel-2 data. The second shows how pyjeo can be imported within an interactive platform for image analysis and visualization. Using an innovative mechanism that interprets Python code within a C++ library on-the-fly, users can benefit from all functions in the pyjeo package. Images are processed in deferred mode, which is ideal for prototyping new algorithms on geospatial data, and assess the suitability of the results created on the fly at any scale and location.

**Keywords:** open-source software; geospatial data; image processing

---

## 1. Introduction

In November 2018, the European Commission (EC) launched the European Open Science Cloud (EOSC). Its aim is to provide a trusted environment for sharing and analyzing data from all publicly funded research. As the science and knowledge service of the EC, the Joint Research Centre (JRC) adopts the principles of open science. Its main task is to support the European Union (EU) policies with independent scientific evidence throughout the whole (European) policy cycle. As an example, the JRC follows the principle that data and other digital objects created by and used for research need to be findable, accessible, interoperable and reusable (FAIR). Indeed, open data plays a central role in its aim to make research more efficient, reliable, collaborative and transparent. In particular, within the geospatial domain, there has been a major movement towards open data. With the launch of the Sentinel satellites of the Copernicus program of the EC, free and fully open geospatial data are available at an unprecedented scale.

However, open data do not necessarily lead to open science. The processing of the data and combination of data and software are hampered by various obstacles such limited interoperability at the semantic level, data discovering capabilities, and the size of data sets [1]. The amount of data becomes difficult to store and process on a personal computer. The Copernicus sensors produce approximately 15 TB of Earth Observation (EO) data per day. Scientists increasingly must rely on

cloud services that provide scalable computing power, where processing can be performed close to the data. Popular examples from the United States are Amazon Web Services (AWS) and Google Earth Engine (GEE). In Europe, an initiative has been launched by the EC, known as the Copernicus Data and Information Access Services (DIAS), to facilitate access to Copernicus data and allow for a scalable computing environment. A key factor for the success of the above-mentioned cloud services is the availability of software tools for the analysis of these big geospatial data.

An important aspect of open science is the transparency and reproducibility of results, irrespective of the platform on which the data are processed. Free and open-source software (FOSS) can play an important role in this aspect. An increasing number of scientific journals move research communities towards a norm where all code is freely available [2]. For instance, for work where novel computer code was developed, authors publishing in this journal should release the code either by depositing in a recognized, public repository or uploading as supplementary information to the publication. Reproducibility can be achieved more easily building on such a culture of openness [2], although the author admits it is only a first step. An increasing amount of research dealing with geospatial data processing has become available in the different domains of geographical information systems (GIS) [3–5]. Within this spectrum of FOSS, the scientific community has been increasingly focusing on those software libraries that provide an application program interface (API). This makes them suitable for batch processing, e.g., when large amounts of data must be processed automatically. By providing an API in a scripting programming language (e.g., Python or R), it becomes relatively easy to use for scientists without a software engineering background. Scripting languages that do not involve a compilation of the code are also suitable for fast prototyping. The lower level algorithmic part of the library, where processing performance is important, is typically written in C or C++. Some software packages combine an integrated graphical user interface (GUI) and an API. The advantage is that results can be visualized immediately in the GUI [3,5].

In this paper, we propose a new FOSS Python package to analyze geospatial data, pyjeo. Similar to the gdalcubes library presented in [6], it supports the concept of on-the-fly data cubes [7]. Geospatial data scientist interested in time series analysis combine overlapping geospatial data that have been acquired at different times. One of the aims of data cubes is to facilitate this time series analysis.

One of the appealing aspects of this concept is that it avoids data duplication and loss of information due to resampling after selecting different resolutions and spatial reference systems [6]. In addition, pyjeo provides a wide variety of image processing algorithms, with a focus on remote sensing image analysis.

The pyjeo library is integrated in two main components of the JRC Earth Observation Data and Processing Platform (JEODPP [8]): jeo-batch and JEO-lab. The first is used for large-scale processing within a commodity cluster computing environment. The second is an interactive platform for image analysis and visualization. The JEO-lab supports deferred processing, in which only those pixels are processed that are actually selected by the interactive viewer at the resolution requested.

In Section 2 the design of pyjeo is presented. We explain how the core code written in C/C++ was bound to Python. This section also provides details of the data model used and how pyjeo fits in JEODPP. In Section 3 we present two use cases on how pyjeo can contribute to open science. The first use case illustrates pyjeo as part of a high throughput computing environment. The second use case shows how pyjeo code can be executed in JEO-lab to design new algorithms interactively using deferred processing.

## 2. Design of pyjeo

The pyjeo library has several dependencies that are all FOSS. It uses the Geospatial Data Abstraction Software Library (GDAL) (https://gdal.org, accessed on 16 October 2019) to access raster and vector data in different formats. In addition, GDAL provides a myriad of functions. In particular, pyjeo uses GDAL to re-project raster images, produce polygon feature layers from raster images, and burn vector geometries into a raster image. The GNU scientific library (GSL) (https:

//www.gnu.org/software/gsl/), accessed on 16 October 2019) is used for its statistical functions and domain transformations (e.g., wavelet transform). Supervised classification algorithms are based on the fast artificial neural network library (FANN) [9] and libsvm [10] (support vector machines). Documentation is written (in-source) in a markup language and formatted to hypertext markup language (HTML) using the documentation generator Sphinx. An alpha version of pyjeo is currently released to be tested by scientists within the JRC. The first release as open source under the European Union Public License (EUPL) is under progress.

## 2.1. From C/C++ to Python

At its core, pyjeo is based on the jiplib library that is implemented in C/C++. This library finds its origin in the open-source pktools software suite for processing geospatial data [11], following a complete re-design including the introduction of a new data model (see also Section 2.2). The jiplib library was then bound to Python and further extended with new developments under the name of a new package, pyjeo. Parallel computing is introduced using the open-source Message Passing Library (openMPI).

The two basic classes that deal with datasets are *Jim* for raster data and *JimVect* for vector data. A schematic overview is shown in Figures 1 and 2 respectively. A container class *JimList* collects instances of the *Jim* class and inherits all methods from the Python list data type (e.g., *append*, *insert*, *remove*, *pop*).

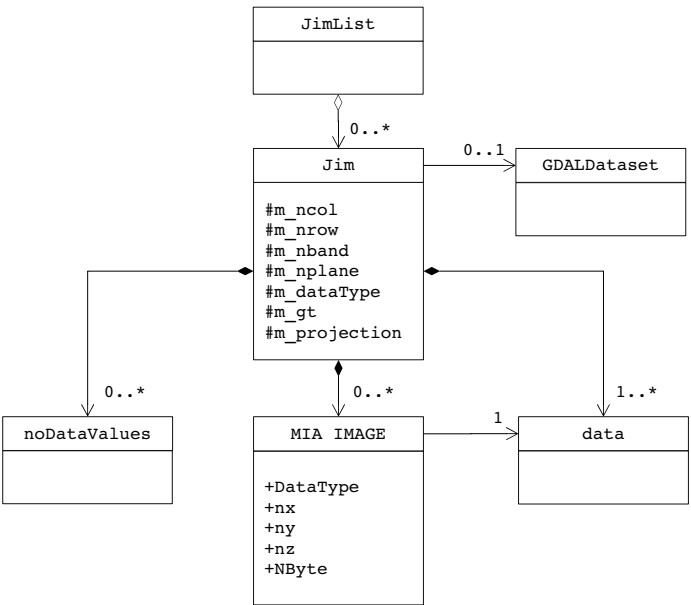

**Figure 1.** Class diagram for raster datasets.

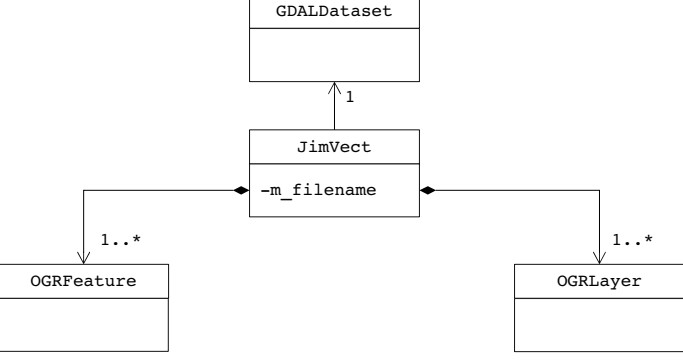

**Figure 2.** Class diagram for vector datasets.

The attribute of the type GDALDataset in *Jim* (Figure 1) is set when a file is read or written. For intermediate results or images created in memory this attribute is not set. For image dimensions, data type, no data values, and coordinate reference system, member attributes are used instead. A special member attribute is used as a bridge to the mialib library. This C library has a focus on morphological image analysis including hierarchical image segmentation based on constrained connectivity [12]. As for the GDALDataset attribute, it is set when needed (i.e., when a function of mialib is called). Both jiplib and mialib can share a data pointer, which is a general-purpose C pointer variable (see also Section 2.2).

In contrast to the *Jim* class, the GDALDataset in *JimVect* (Figure 1) is always set. The *JimVect* instance cannot exist without a GDALDataset. All features are read into memory when opening a dataset to provide fast access. Both the features and layers are member attributes from *JimVect* and are types from the GDAL library (*OGRFeature* and *OGRLayer*).

Python bindings to the C/C++ library are made available through the Simplified Wrapper and Interface Generator (SWIG [13]). This generator produces a wrapper file that is compiled and linked into a dynamic library (see Figure 3). SWIG can bind C and C++ with several scripting languages, among which Python and R.

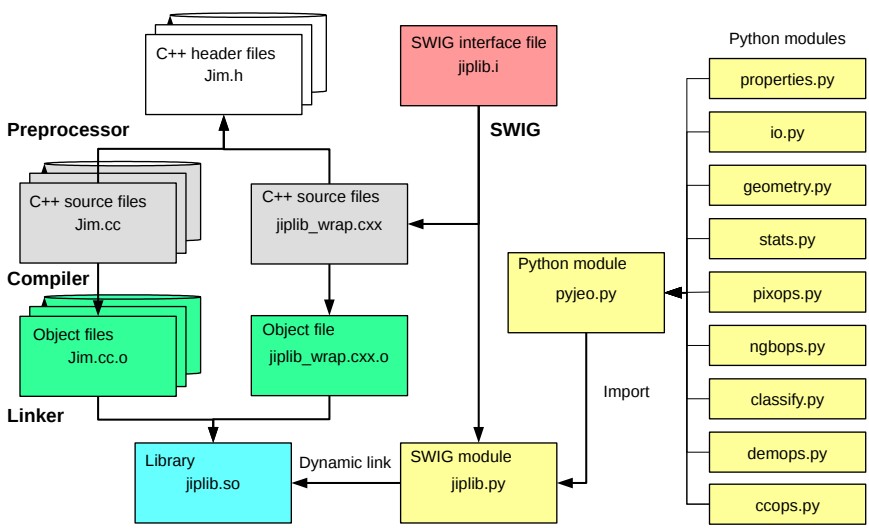

**Figure 3.** Using SWIG to build a Python interface from C++. Header files in white, C/C++ source files in gray, object files in green, dynamic library in cyan, Python modules in yellow.

The main input file for SWIG is the interface file (*jiplib.i*). This file lists all the functions of the C/C++ jiplib library that need to be bound to the target scripting language. Any customization to the default behavior of the mapped functions and their arguments can also be made in the interface file. SWIG generates a wrapper (C++ source) file (*jiplib_wrap.cxx*) that is be compiled and linked with the source and object files of the jiplib C/C++ library. The result is a library (*jiplib.so*) and a package in the target programming language (*jiplib.py*). This Python package provides access to the functions listed in (*jiplib.i*). For classes originating from the C/C++ library, SWIG creates a proxy class in the target language. This allows users to create objects of the original classes directly from Python.

To improve the user interface, a new Python package *pyjeo.py* is created manually on top of the automatically generated package *jiplib.py*. The pyjeo package organizes the available functions in the Python modules as shown in Figure 3 (right column). It also includes new extensions to the library, written in Python.

The first module, **properties**, is a general module to get and set properties of the basic classes *Jim* (for raster datasets) and *JimVect* (for vector datasets). For example, the method *properties.getBBox()* returns the bounding box in the spatial reference system of an instance of the *Jim* class. Input and output operations such as reading and writing datasets from and to disk are covered in the **io** module. The **geometry** module groups operations that relate to the geometry of a dataset. Subsetting data cubes, aggregating raster pixels based on vector feature overlays (i.e., zonal statistics), and warping are some examples. Statistical functions such as minimum, maximum, means, and histograms are part of the **stats** module. Algorithmic functions based on pixel operations and neighborhood operations are grouped in the modules **pixops** and **ngbops** respectively. Typical pixel operations are: convert the data type, threshold and mask, and pixel wise arithmetic. Many of these functions can also be called directly on the raster object (Jim) using operator overloading:

$$jim1 + = jim2 \quad \text{(in-place addition)} \tag{1}$$

$$jim3 = jim1 + jim2 \quad \text{(addition returning a new object)} \tag{2}$$

Example of a threshold operation to set all pixel values in *jim1* less than 100 to 0 (in-place):

$$jim1[jim1 < 100] = 0 \tag{3}$$

Examples of neighborhood operations include convolutional and morphological filtering. The **classify** module include multilayer artificial neural networks. Both fully connected and sparsely connected networks are implemented. Also implemented are support vector machines and symbolic machine learning (SML [14]). Other classification algorithms can easily be integrated using the bridge to other data models (see also Section 2.2). This is also valid for algorithms not related to classification. Operations that are specific to digital elevation models can be called from the **demops** module: e.g., calculation of slope (directions), pit removal [15,16], and contribution of drainage areas [17]. Finally, the **ccops** module supports several connected-component operations including image segmentation algorithms such as watersheds and constrained connectivity [18] and the calculation of different distance (e.g., Euclidean and Geodesic) measures.

### 2.2. Data model

Most popular geospatial raster data file formats support only two-dimensional (2D) image data. Several of them, including the popular GeoTIFF, support stacks of 2D images, often referred to as bands. Only a few support multi-dimensional data, for instance the Network Common Data Form (NetCDF [19]). Only recently, GDAL has generalized the traditional 2D raster data model, to address higher dimension datasets from its API (from GDAL v.3.1). The data model used in pyjeo is a multi-band three-dimensional (3D) model, where each band represents a 3D contiguous array in C/C++ of a generic data type using a general-purpose C pointer variable (void). Two dimensions refer to the spatial domain (x and y) and pyjeo refers to the third dimension as "plane" (Figure 4). This plane is typically used for either the temporal or spectral dimension, but can also be used to address volumetric data.

When reading a multi-band raster dataset into a Jim raster data object, the user can choose if the bands are to be considered to be planes, resulting in a single-band 3D object, or as bands (resulting in a multi-band 2D raster object). Planes and bands can be stacked (*stackPlane*, *stackBand*) and subset (*cropPlane*, *cropBand*) as shown if Figure 4. Dimensions must correspond across all bands within the same *Jim* object. To collect objects with different dimensions, a *JimList* can be used. Multi-dimensional data with dimensions above three are not supported in pyjeo. However, most (other than arithmetic) functions in practice operate on 1D (e.g., spectral filtering), 2D (e.g., spatial filtering). Few functions in pyjeo operate on the 3D data. Functions on the individual bands can easily be processed in parallel as different data pointers are used for each band.

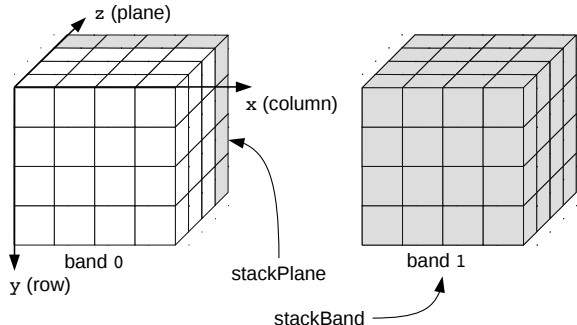

**Figure 4.** The pyjeo raster data model is a multi-band 3D dataset, represented by the Python class *Jim*. The function stackPlane adds a plane to the dataset, whereas stackBand adds a new band with the same dimension as the existing bands.

The simple data model with contiguous arrays allows combination of pyjeo with other software such as GDAL, GSL, and Python packages that are compatible with NumPy arrays (e.g., SciPy). A direct bridge to NumPy arrays is included in pyjeo, without the generation of an extra copy in memory. In particular, when dealing with geospatial data that have a large memory footprint, careful memory handling is important. To this end, users can choose if pyjeo functions modify objects in-place or return a new object. Modifying objects in-place can minimize the memory footprint, but the original object is lost. According to the principle of command-query-separation [20], functions that change state should not return values and functions that return values should not change state. Following this principle, pyjeo functions that modify the object in-place return None. To avoid confusion with functions that return a new object, they have been implemented as methods that are bound to a class. For instance, to subset the first band of a *Jim* instance in-place, use:

$$jim.geometry.cropBand(0) \tag{4}$$

After the call, *jim* will be a single-band raster dataset. No new object is returned, and the original object is lost. A corresponding function is implemented that is not bound to a class:

$$jim0 = pyjeo.geometry.cropBand(jim, 0) \tag{5}$$

returns the new single-band object *jim*0. In this case, all objects are passed explicitly as arguments and are not modified.

### 2.3. Integrating pyjeo for Big Data Analytics

Within the Big Data Analytics project of the JRC, a versatile data-intensive computing platform has been developed. It accommodates different service levels ranging from large-scale batch processing to interactive visualization and analysis of geospatial data. Referred to as the JRC Earth Observation Data and Processing Platform (JEODPP [8]), it currently can store 12 Petabytes of data, from which the vast majority are geospatial data originating from the Copernicus program. All data are made available to 1500 processing cores via 10 Gb/s switches.

A PostGIS database that is specific to the JEODPP defines all available geospatial data, describing them as data collections. The data are stored in their original format, resolution and spatial reference system. A representational state transfer (RESTful) web service is implemented that can handle hypertext transfer protocol (HTTP) requests to query the collections in the data base. Instead of dealing with individual file paths, users of the platform should be able to work with collections of data and construct on-the-fly data cubes. Because the pyjeo library is agnostic with respect this database, a thin layer with an extra class *Collection* was created. This layer was designed based on the concept of openEO [21] (https://openeo.org/, accessed on 16 October 2019). It aims at a common, open-source

interface between Earth observation data infrastructures and user applications. The objective is to allow users to query cloud-based back-ends and carry out computations on them in a simple and interoperable way.

Instances of the *Collection* class can be filtered to select specific datasets from the archive. The filters query the RESTful web service. A cube can be created on-the-fly in a target reference system. Re-projection, resampling, and composition occur, if needed, at the time of the data cube creation. An example of Python code that creates a data cube on-the-fly from a collection is given in Listing 1 (lines 4–8). An advantage of the extra layer is that the pyjeo library is well separated from the database. The library can still be used on individual files on a local file system. By adapting the implementation of the *Collection* class, pyjeo can also be used in a cloud service that has its own database and proprietary API. The data cube is created in memory. While it is desirable for performance and to avoid creating intermediate files, it can also be a limitation when dealing with large imagery. Within the JEODPP this issue is addressed by two components that correspond to different use cases.

The first component, jeo-batch, is used to process data at large scale in batch mode. It has access to a processing cluster of 1500 cores. Job orchestration is performed with HTCondor [22], which has a focus on high throughput computing. Most image processing tasks have indeed little or no concurrent intercommunication that is typical for this type of computing. Users can import the pyjeo library in their Python processing scripts. If needed, images can be tiled to fit into the available memory. Notice that many image collections, including Sentinel-2 from Copernicus, already are provided in a tiled format. The $100 \times 100$ km tiles follow the Military Grid Reference System (MGRS) and are projected in Universal Transverse Mercator (UTM). The tiles, even the bands at the finest spatial resolution of 10 m, easily fit in the minimum of 10 GBytes of memory of a single core of the JEODPP jeo-batch infrastructure.

The second component, JEO-lab, provides a JupyterLab environment for interactive image analysis and visualization [23]. It is based on a C++ library that implements a dynamic tile server. This server provides raster tiles to the iPyleaflet client in the web map tile server (WMTS) HTTP protocol. These tiles are then visualized in deferred mode on a dynamic map. An interesting feature has been implemented in the JEO-lab library that allows users to add an image processing function, written in Python, inside a notebook of the JupyterLab environment. By importing the pyjeo module, the full set of features from this module are also available. The Python function returns an image object that is then passed to the subsequent steps of the processing chain needed to visualize the result on the interactive map. The integration of Python code within a C++ library is enabled by a special Python module *inspect*. The Python code is read and sent to the C++ tile server. This tile server initiates a Python interpreter on-the-fly. The interpreter executes the Python code for each tile request. The C++ code takes care of passing the input image to the Python code and reading the returned image. This feature adds a lot of flexibility to JEO-lab and enables fast prototyping of complex algorithms in view of porting them to jeo-batch for large-scale processing.

## 3. Use Cases

To illustrate how the pyjeo library can contribute to open science, two use cases are presented. In particular, in the first use case pyjeo is applied for a high throughput computing task within jeo-batch, the component of JEODPP that focuses on large-scale processing. The second use case shows how pyjeo is integrated within the JEO-lab, the interactive analysis and visualization component of JEODPP.

### 3.1. Large-Scale Processing with pyjeo in Jeo-Batch

A global cloud free composite was created over land, based on atmospherically corrected Sentinel-2 images [24]. The images used were acquired between January 2016 to September 2017. The composition was performed in two steps. The first step was to select a minimum set of images from which a cloud free composite could be obtained. The full set of images was not available in the JEODPP. Instead, the selection was based on metadata that were downloaded from the Copernicus Open Access

Hub (https://scihub.copernicus.eu/, accessed on 16 October 2019), which provides complete, free and open access to Sentinel-1, Sentinel-2, Sentinel-3 and Sentinel-5P user products. These meta data include a cloud mask in vector format and a quicklook image. The quicklooks represent a lossy compressed spatial and spectral subset of the acquired image. The optimization algorithm is explained in more detail [25] (an open access journal).

The second step was to download the Sentinel-2 images at full spatial and spectral resolution. Images were provided at processing level 1C, which includes radiometric and geometric corrections. Atmospheric correction was performed on the JEODPP using Sen2Cor [26] software. The composite was then calculated using pyjeo, based on a combination of rules. A first rule is based on the class code in the scene classification (SCL). This thematic layer, produced by the Sen2Cor software, labels the pixel in 12 classes. The 12 classes were ranked according to their relevance to the global composite (see Table 1). Pixels coded as vegetation (SCL code 4) received the highest priority (rank 0), whereas no data (SCL code 0) received the lowest priority (rank 11).

**Table 1.** Ranking of the scene classification used for the composite rule.

| Rank | Description | SCL Code |
| --- | --- | --- |
| 0 | Vegetation | 4 |
| 1 | Bare Soils | 5 |
| 2 | Water | 6 |
| 3 | Dark Area Pixels | 2 |
| 4 | Snow/Ice | 11 |
| 5 | Cirrus | 10 |
| 6 | Cloud Shadows | 3 |
| 7 | Clouds low probability/Unclassified | 7 |
| 8 | Clouds medium probability | 8 |
| 9 | Clouds high probability | 9 |
| 10 | Saturated/Defective | 1 |
| 11 | No Data | 0 |

The code to composite a single MGRS tile was implemented in Python using the pyjeo library (see Listing 2). The algorithm iterates through the minimum set of images, based on their unique identifiers that are provided as input (*productlist*). A collection is created and filtered based on the product id (line 5). After selecting the spectral bands of interest including the SCL band (lines 6–7), the collection is loaded into a data cube, with a spatial resolution of 10 m (line 8). A nearest neighbor (default) resampling is performed for the SCL band, as it was created by Sen2Cor at a spatial resolution of 20 m. In lines 10–11, this band is extracted and re-coded according to the ranking of Table 1. The normalized difference vegetation index (NDVI) is calculated in lines 13–19, based on the red ('B4') and near infrared band ('B8').

During the first iteration, the composite is initialized as the first image in the minimum set (line 24). In the next iterations, it is updated (lines 26–30). Pixels with a lower SCL rank have a higher priority and are always selected with respect to pixels with a higher SCL rank. In case the ranking is the same, the pixels with the maximum NDVI is selected. This rule is not applied for pixels labeled as water (SCL code 4, rank 2) for which NDVI values are not relevant. Finally, the composite is written to file in a compressed and tiled GeoTIFF format (line 32).

Listing 1: Composite algorithm using pyjeo in batch mode

```
1   rank=[4,5,6,2,11,10,3,7,8,9,1,0]
2   composite=None
3   for productid in productlist:
4   coll=Collection()
5   coll.filterOn('id',productid)
6   bands=['B2','B3','B4','B8','SCL']
7   coll.filter_bands(bands)
8   jim=coll.load_collection(resolution={'spatial':[10,10]},otype='UInt16')
9
10  scl=pj.geometry.cropBand(jim,jim.properties.nrOfBand()-1)
11  scl.classify.reclass(classes=rank,reclasses=range(0,len(rank)))
12
13  red=pj.geometry.cropBand(jim,2)
14  red.pixops.convert('Float32')
15  nir=pj.geometry.cropBand(jim,3)
16  nir.pixops.convert('Float32')
17  ndvi=nir-red
18  ndvi/= nir+red
19  ndvi[nir+red<0.1]=-1
20
21  if composite is None:
22  ndvireduced=pj.Jim(ndvi)
23  sclreduced=pj.Jim(scl)
24  composite=pj.Jim(jim)
25  else:
26  index=(sclreduced>scl)
27  index|=((sclreduced>=scl) & (ndvireduced<ndvi) & (scl!=2))
28  ndvireduced[index]=ndvi
29  sclreduced[index]=scl
30  composite[index]=jim
31
32  composite.io.write('composite.tif', co=['COMPRESS=LZW','TILED=YES'])
```

The compositing was performed on the JEODPP cluster on an MGRS tile by tile basis. With no dependencies between the tiles or need for communication between the tasks, this is a typical example of an embarrassingly parallel workload. The processing time for the different tasks to create the global cloud free composite is shown in Table 2.

**Table 2.** Throughput and total processing time to create world composite (using 952 cores).

| Task | Throughput | Total Processing Time |
|------|-----------|----------------------|
| selection | 1.5 tiles/hour/core | 20 h |
| Sen2Cor | 0.6 tiles/hour/core | 50 h |
| compositing | 2 tiles/hour/core | 15 h |

Once the composites were created, a virtual dataset (VRT) with overview files was created for all the composites within the same UTM zone. From this VRT, a new collection was created. This allows the global composite to be visualized interactively in JEO-lab, as shown in Figure 5. In the next section, it will be shown how pyjeo can be integrated in an interactive analysis environment.

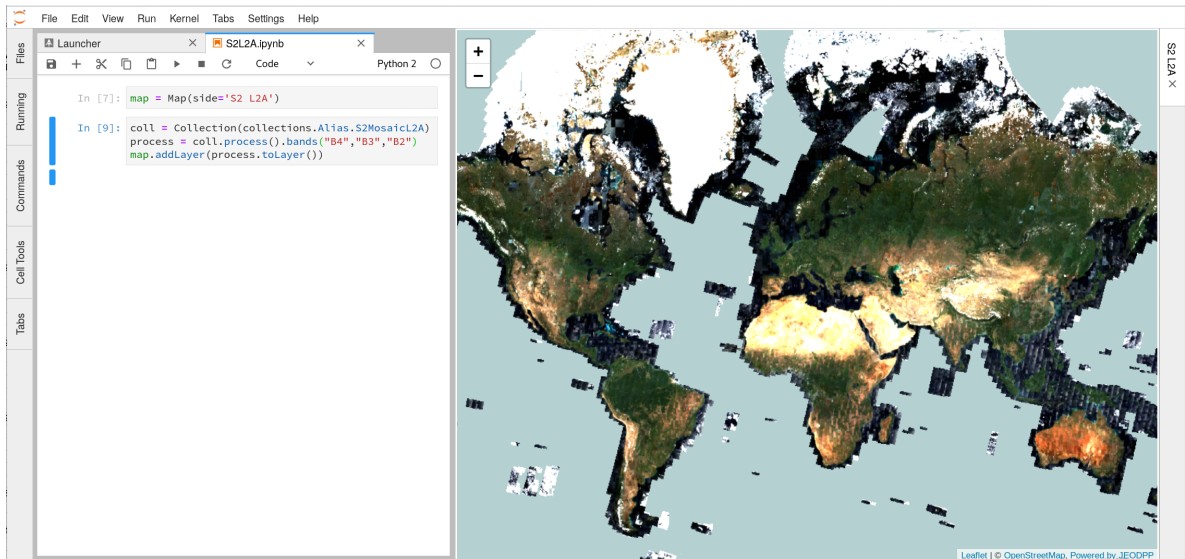

**Figure 5.** Sentinel-2 global composite based on a minimum set of overlapping images visualized from a JupyterLab notebook in JEO-lab.

### 3.2. Interactive Processing with Pyjeo in JEO-Lab

Prototyping algorithms for products over large areas such as in Section 3.1 can be tedious. One of the main concerns is the ability of the algorithm to generalize. Parameters that work well for one area might fail for another. A processing cluster can speed up the typical iterative approach of designing a new algorithm to get insight in its global behavior. With the Google Earth Engine (GEE [27]) a new way of prototyping was introduced. Users do no longer need to process the entire dataset at each iteration. Instead, they pan and zoom to an area of interest and evaluate the results on-the-fly using deferred processing. Inspired by the GEE, a similar approach has been implemented in JEO-lab [8,28], the interactive analysis and visualization platform within the JEODPP.

In Section 3.1, a global cloud free composite was presented as the result of a large-scale processing job in jeo-batch. In Figure 5, it was also shown how JEO-lab can be used to visualize the results interactively. Users can smoothly visualize the result for any position on Earth and at any resolution. The deferred processing only loads a subset of files at the corresponding overviews. The same mechanism can be used to execute arbitrary Python code, harnessed with the full set of features available in pyjeo.

Suppose we want to skip the selection of the minimum set described in [25] followed by the composition algorithm described in Listing 2. Instead, we aim at creating a composite based on the five least cloudy overlapping images for some location and acquisition time window. We also replaced the compositing scheme used in Section 3.1 with a new rule that is based on the maximum Euclidean distance to clouds [29]. Pixels risk to have contaminated reflectance values due to neighboring clouds. This can be due to partial cloud cover or multiple scattering in the atmosphere. Selecting pixels with a larger distance to clouds mitigates this effect. Other metrics such as the distance between a reference date and the acquisition date can also be combined in a weighted scheme [30].

The new code is executed in deferred mode, which allows the designer of the algorithm to analyze results on-the-fly. Once fine-tuned, the code can be run in jeo-batch as presented in Section 3.1.

The code in Python on the left pane of Figure 6 cover the complete processing chain from filtering the collection, composting and creation of the interactive map.

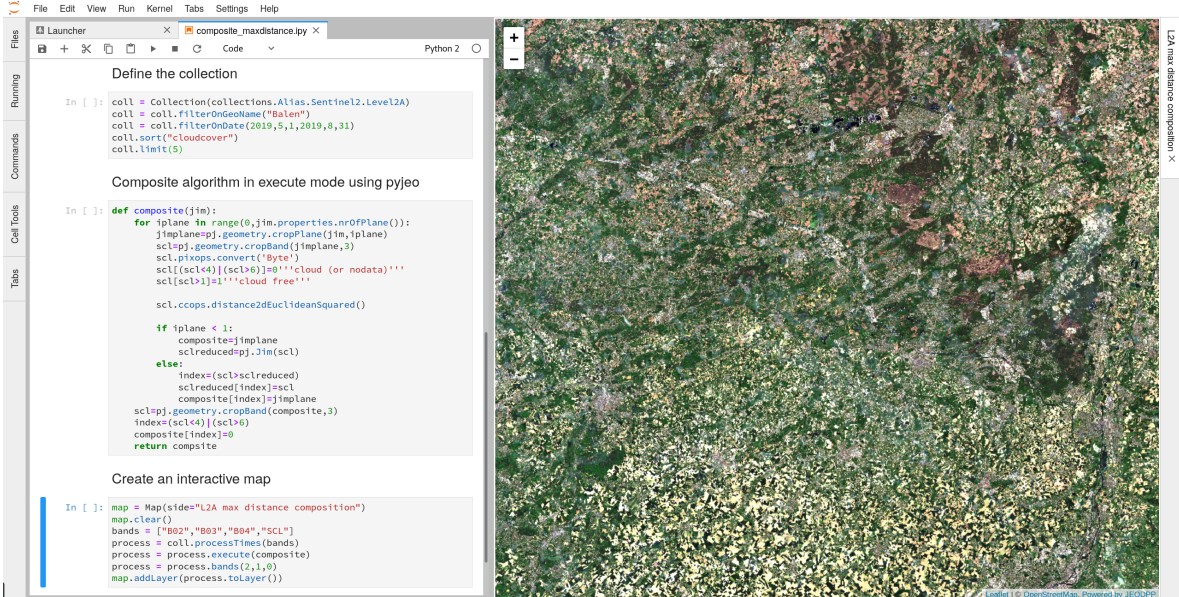

**Figure 6.** Interactive analysis and visualization in JEO-lab. The definition of the composition function is written in Python using pyjeo functions (left). The execution is performed in deferred processing, where each tile is processed in parallel at the extent and scale the user has visualized in the interactive map viewer (right).

Filtering a collection in JEO-lab is very similar to jeo-batch (Listing 1, lines 4–7). In the previous use case, the collection was filtered based on the product id. Here, the collection is filtered on location and date. The five least cloudy images are selected with a consecutive sort and limit operation. In JEO-lab, all images are projected on-the-fly in a web (pseudo) Mercator projection (EPSG:3857). The resulting temporal stack of images is therefore not loaded in the original UTM projection as in jeo-batch. Another difference with jeo-batch is that the bands in JEO-lab are not filtered from the collection object. Instead they are defined in the process object from which the composite function will be called.

The function that implements the composite algorithm is shown in Listing 2. The input is a *Jim* object that contains the temporal stack as planes and the selected bands following the design in Figure 4. Similar to the implementation in jeo-batch, the cloud free composite is created incrementally in this example. At each iteration, the next plane is selected (line 3). The cloud information is extracted from the Sentinel-2 level 2A scene classification (SCL) band (line 4). Using a simple threshold operator, a binary mask is created (lines 5–7) on which the Euclidean distance transform is calculated. We consider the SCL code 4 (vegetation), 5 (bare soils) and 6 (water) as valid pixels for the composite. These pixels obtain a value 1 in the binary mask (line 7). All pixels with a different code will receive a value 0 in the binary mask (line 6). if the current plane is cloud free, it is selected as the composite and no further images are considered (lines 9–11). The Euclidean (squared) distance from a cloud free pixel (value 1) to a cloudy pixel (value 0) is calculated in line 13. Updating the composite based on the maximum distance is performed in lines 15–21. The distance is calculated in-place and is obtained from the object *scl*. At the end of the update, the SCL is extracted from the composite (line 23). It is used to set all remaining cloudy pixels to value 0 (lines 24–25), which makes these pixels transparent in the interactive map.

Listing 2: Composite algorithm using pyjeo

```
1  def composite(jim):
2  for iplane in range(0,jim.properties.nrOfPlane()):
3  jimplane=pj.geometry.cropPlane(jim,iplane)
4  scl=pj.geometry.cropBand(jimplane,3)
5  scl.pixops.convert("Byte")
6  scl[(scl<4)|(scl>6)]=0'''cloud (or nodata)'''
7  scl[scl>1]=1'''cloud free'''
8
9  if scl.stats.getStats(["min"])["min"]>=1:
10 composite=jimplane
11 break
12
13 scl.ccops.distance2dEuclideanSquared()
14
15 if iplane < 1:
16 composite=jimplane
17 sclreduced=pj.Jim(scl)
18 else:
19 index=(scl>sclreduced)
20 sclreduced[index]=scl
21 composite[index]=jimplane
22
23 scl=pj.geometry.cropBand(composite,3)
24 index=(scl<4)|(scl>6)
25 composite[index]=0
26 return composite
```

A new map object for visualizing the processed images is created. The deferred processing is based on a process object that is created from the filtered collection object. The bands are also selected here. The execution of the composite function is then called. The bands are re-ordered to obtain a true color image (red, green, blue). Adding the process to the map as a layer triggers the deferred processing.

Using a different mapping scheme between the scene classification and the binary mask is one example of how the algorithm can be fine-tuned on-the-fly. More subtle tuning might be obtained by using weights for the different cloud probabilities when calculating the distance metric.

## 4. Conclusions

A new Python package, pyjeo, was presented for analyzing geospatial data. It offers a variety of functions that are grouped in different Python modules. The selected data model holds the data to be processed in memory. This is desirable for performance and it avoids generating intermediate files. Users can use a direct bridge from an instance of Jim, the raster data class in pyjeo, to a NumPy array. This enriches pyjeo with a plethora of new functions that originate from other packages that use this data array model (e.g., SciPy). Data are hereby shared in memory, without the need for an extra copy. Another technique applied in pyjeo to reduce the memory footprint is the dual design of functions. Most functions on raster data in pyjeo are available both as in-place methods bound to objects and functions that return a new object and leave their arguments unchanged. An increasingly popular concept in geospatial data science is the creation of on-the-fly data cubes from data collections. This typically involves a database that registers the available datasets as data collections. This concept is also supported by the data model of pyjeo. A RESTful API, dedicated to a database referring to available datasets and their associated metadata, handles queries that filter collections in time and space or any of their attributes. The filtered collection can be loaded into data cubes and further processed with pyjeo.

Two use cases were presented to illustrate how pyjeo can contribute to open science. With the free, full, and open data policy of the Copernicus program of the European Union, the description of the

underlying algorithms, and the use of free and open-source software, both use cases address indeed the main principles of open science: transparency and reproducibility of results. Free and open public access to the code is foreseen in the near future by releasing it under EUPL, a free software license that has been created and approved by the European Commission. The EUPL license is compatible with many common FOSS licenses, including GPL (v.2 and v.3), AGPL (v.3), and CeCILL (v.2.0 and v.2.1). The open-source release of pyjeo will bring in new contributors, increase code robustness through a wider user community, promote collaboration beyond the JRC, and enable the integration of the software into other infrastructures.

**Author Contributions:** P.K. was responsible for writing the initial version of the paper and is the main contributor of jiplib library. O.P. contributed to the writing of the pyjeo modules during his traineeship at JRC from mid-July to mid-December 2018. D.D.M. enabled the interactive execution of pyjeo and other Python packages within the JEO-lab environment. P.S. ideated the need for exposing functions previously written in C and C++ to the Python language and contributed to the writing of all software components as well as the review and editing of the paper. All authors contributed to the writing of the paper.

**Funding:** This research received no external funding.

**Conflicts of Interest:** The authors declare no conflict of interest.

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
