# Peer review of "pyjeo: A Python Package for the Analysis of Geospatial Data"

_ijgi, doi:10.3390/ijgi8100461_

Round 1
Reviewer 1 Report
I found the paper interesting and I look forward to testing the beta release of this python package. Currently, there are a lot of studies where this package can use used. The paper is well described especially the data model section and the integration for big data analytics. One can clearly understand the relevance of this FOSS package in open science, through the use cases. Overall I found an interesting contribution to the GIScience and FOSS communities.
Author Response
Thank you for this positive feedback.
Reviewer 2 Report
In this paper, a pre-release of a new FOSS Python package, pyjeo, is presented. The package promises to bring advanced functionalities into large scale spatial data processing (raster, mainly) optimized for a distributed computing environment.
The topic of the paper is of the utmost relevance to both present and future GI science. The detailed model specification presented by the authors provides with a good level of understanding of the work. The paper is technically sound and the quality of English is good. However, some text styling/formatting issue should be fixed. Below, the comments.
Once the acronym is spelled out at the first writing, use the acronym without repeating the full spelling - see e.g. line 24 Reference in text: please, place references outside the acronyms brackets (e.g. Full Name (FL) [1])- see e.g. lines 77, 79 Refer to figures, sections, tables, listing, etc by capitalizing the first letter and using the entire name with no abbreviations (e.g. Figure 1, Section 1, Listing 1, etc) Cite all the reported equations in the text.
Below, some suggestion for improving the Conclusions section, which should be extended to include considerations not only on the pure technical content in my opinion.
Add a couple of statements to connect the presented use cases with open science principles (or remark the main concepts) About the software: To date, everything is internal to the JRC infrastructure. Explain briefly the release planning and how the FOSS community (users + devs) would get advantage from this new library. A good point to spell out would be whereas EUPL is compatible with most of the traditional open-source licenses (e.g. GNU) and if this favor future integrations into other infrastructures/systems, etc.
After the above changes, I believe the paper will be ready for the publication.
Author Response
Thank you for these suggestions, which have been taken into account. A description of the release as open source including the license has been added. Following your suggestion on how to improve the conclusions section, we added the following paragraph:
Two use cases were presented to illustrate how pyjeo can contribute to open science. With the free, full, and open data policy of the Copernicus programme of the European Union, the description of the underlying algorithms, and the use of free and open source software, both use cases address indeed the main principles of open science: transparency and reproducibility of results. Free and open public access to the code is foreseen in the near future by releasing it under EUPL, a free software license that has been created and approved by the European Commission. The EUPL license is compatible with many common FOSS licenses, including GPL (v.2 and v.3), AGPL (v.3), and CeCILL (v.2.0 and v.2.1). The open source release of pyjeo will bring in new contributors, increase code robustness through a wider user community, promote collaboration beyond the JRC, and enable the integration of the software into other infrastructures.
Reviewer 3 Report
This is a very interesting article on a very decent and important topic. I enjoyed reading the work and have minor comments for improvement:
The introduction section can be further improved by addressing and stressing the importance and value of Open Science in general and for the GI research in particular. There are several articles on this that you might be interested to read such as: Bakillah, M., & Liang, S. (2016). Open geospatial data, software and standards. I would further suggest to present some other examples of research in Open Source GIS to stress the importance of it in these days. Again there are several examples of OS GIS tools/services for different application domains such as:- Rupnik, E., Daakir, M., & Deseilligny, M. P. (2017). MicMac–a free, open-source solution for photogrammetry. Open Geospatial Data, Software and Standards, 2(1), 14. - Grizonnet, M., Michel, J., Poughon, V., Inglada, J., Savinaud, M., & Cresson, R. (2017). Orfeo ToolBox: Open source processing of remote sensing images.
Open Geospatial Data, Software and Standards
,2
(1), 15. - Furthermore, I think the use cases section could be revised and improved before getting published. The content is fine and there is no negative points about it. It's just that the story-telling and the way it's presented can be enhanced. Perhaps the authors can read the two examples/references I provided above and get some hints from them on how the scenarios and cases studies can be better represented. However, this is just a minor suggestion, and if the authors are happy with it's current version I would still support it's publication. Good luck
Author Response
We appreciate this positive feedback. Other examples of research have been added, including the references from Bakillah and Rupnik. With respect to the suggestion on the enhancement of the case study, we believe this is indeed a matter of taste. Given the fact that all reviewers (including this reviewer) did not object to keeping this part as is, we preferred not to change it.